# A Novel, Quick, and Reliable Smartphone-Based Method for Serum PSA Quantification: Original Design of a Portable Microfluidic Immunosensor-Based System

**DOI:** 10.3390/cancers14184483

**Published:** 2022-09-16

**Authors:** Francisco Gabriel Ortega, Germán E. Gómez, Coral González-Martinez, Teresa Valero, José Expósito-Hernández, Ignacio Puche, Alba Rodriguez-Martinez, María José Serrano, José Antonio Lorente, Martín A. Fernández-Baldo

**Affiliations:** 1IBS Granada, Instituto de Investigación Biosanitaria de Granada, 18012 Granada, Spain; 2GENYO, Centre for Genomics and Oncological Research, Pfizer/University of Granada/Andalusian Regional Government, PTS Granada, Avenida de la Ilustración 114, 18016 Granada, Spain; 3Instituto de Investigaciones en Tecnología Química (INTEQUI), Departamento de Química, Universidad Nacional de San Luis, CONICET, Chacabuco y Pedernera, San Luis D5700BWS, Argentina; 4Department of Medicinal and Organic Chemistry, School of Pharmacy, University of Granada, Campus Cartuja s/n, 18071 Granada, Spain; 5Integral Oncology Division, Virgen de las Nieves University Hospital, 18071 Granada, Spain; 6Department of Pathological Anatomy, School of Medicine, University of Granada, 18071 Granada, Spain; 7Lab. of Genetic Identification, Department of Legal Medicine & Toxicology and Physical Anthropology, School of Medicine, University of Granada, 18016 Granada, Spain; 8INQUISAL, Departamento de Química, Universidad Nacional de San Luis, CONICET, Chacabuco 917, San Luis D5700BWS, Argentina

**Keywords:** PSA, cancer biomarker, liquid biopsy, cancer diagnosis, magnetic microbeads, microfluidic immunosensor

## Abstract

**Simple Summary:**

Prostate cancer (PCa) is the most frequently diagnosed malignancy and second most common cause of cancer-related death in males. An early diagnosis is crucial to improve the prognosis. Prostate-Specific Antigen (PSA) is the most widely used biomarker for PCa, but this type of biomarker analysis is performed in centralized laboratories, delaying the diagnosis and initiation of treatment. Our team has developed a miniaturized platform for portable PSA quantification to overcome this shortcoming. It includes a microfluidic chip, immune capture of PSA by magnetic microbeads, and electrochemical quantification. The utilization of a micro-potentiostat allows PSA levels to be read on a smartphone in less than 30 min. This technique was found to offer a fast, easy, specific, sensitive, and reproducible method for PSA quantification. Further research is warranted to verify these findings and explore its potential application at all health care levels.

**Abstract:**

We describe a versatile, portable, and simple platform that includes a microfluidic electrochemical immunosensor for prostate-specific antigen (PSA) detection. It is based on the covalent immobilization of the anti-PSA monoclonal antibody on magnetic microbeads retained in the central channel of a microfluidic device. Image flow cytometry and scanning electron microscopy were used to characterize the magnetic microbeads. A direct sandwich immunoassay (with horseradish peroxidase-conjugated PSA antibody) served to quantify the cancer biomarker in serum samples. The enzymatic product was detected at −100 mV by amperometry on sputtered thin-film electrodes. Electrochemical reaction produced a current proportional to the PSA level, with a linear range from 10 pg mL^−1^ to 1500 pg mL^−1^. The sensitivity was demonstrated by a detection limit of 2 pg mL^−1^ and the reproducibility by a coefficient of variation of 6.16%. The clinical performance of this platform was tested in serum samples from patients with prostate cancer (PCa), observing high specificity and full correlation with gold standard determinations. In conclusion, this analytical platform is a promising tool for measuring PSA levels in patients with PCa, offering a high sensitivity and reduced variability. The small platform size and low cost of this quantitative methodology support its suitability for the fast and sensitive analysis of PSA and other circulating biomarkers in patients. Further research is warranted to verify these findings and explore its potential application at all healthcare levels.

## 1. Introduction

Liquid biopsies offer a noninvasive alternative to surgical biopsies [1] and are frequently used to study circulating tumor cells and cell-free DNA in blood samples [2]. They have long been employed to investigate protein biomarkers in venous blood [3], and more than 100 protein biomarkers have been developed over the past few decades for clinical diagnosis and evaluation of the therapeutic response or disease recurrence. The most widespread biomarkers approved by the USA Food and Drug Administration (FDA) and European Medicine Agency (EMA) [4] for urology and prostate cancer (PCa) disease include prostate-specific antigen (PSA) [5], carbohydrate antigen 125 (CA 125) [3], and carcinoembryonic antigen (CEA) [6], among others [7]. Despite the low sensitivity and specificity of these biomarkers, their increased levels in cancer provide clinicians with useful initial information about the disease status of patients [8]. However, the time taken by current techniques to analyze markers produces a delay in the delivery of results to clinicians [9,10]. This study proposes a portable and fast method for in situ biomarker analysis of PSA levels in patients with possible PCa. The protein PSA is specifically synthesized by the prostate, and its production is influenced not only by prostate size and androgen activity but also by prostate inflammation. PSA levels are generally very low in healthy males but elevated in the presence of prostatic disease, and they were found to be increased in 65% of patients with PCa [5]. The measurement of serum PSA levels is currently considered to be the most sensitive test to diagnose and stratify the severity of PCa.

Radioimmune assay (RIA) and enzyme-linked immunosorbent assay (ELISA) are currently used to determine PSA in the clinical setting, while fluorescent immunoassay, photoelectrochemical detection, and electrochemical immunosensors, among others, are used for research purposes [11,12,13,14,15,16,17,18,19,20,21,22,23,24,25,26]. In particular, electrochemical immunosensors have been proposed as a promising method for potential clinical application due to their high sensitivity and specificity [27,28,29,30,31,32]. They offer the possibility of performing simultaneous analyses and are easy to miniaturize, providing a simple analysis technique at a lower cost [33,34,35,36,37]. Immune-electrochemical sensors are highly specific due to the affinity between antibody and antigen, while the electrochemical signal induced by the hybridization of antigen–antibody is the measurable signal that correlates with the protein concentration [33,35]. The coupling of immunosensors with microfluidic systems provides additional advantages, including smaller sample volumes, faster turnaround times, and lower costs [38,39,40,41,42,43]. These systems contain microchannels for transporting fluids and some or all of the components required for an immunoassay [38,39,40,41,42,43]. It has been demonstrated that the combination of microfluidic technology with electrochemical sensing improves the overall detection capability [39,40,43].

Microfluidic immunosensors have shown promise for the analysis of tumoral biomarkers. Various types of particles, such as magnetic microbeads, have recently been incorporated to amplify the reaction surface and sensor response, enhancing the sensitivity of biomarker detection [39,44,45]. The utilization of these nanoparticles as solid support for electrochemical reaction has been proposed for the development of PSA sensors [46,47,48]. Further research is required to design and manufacture this type of biosensor, and there is a need to develop a portable platform that allows in situ analysis in real conditions and not only in the laboratory setting. Further steps to be taken include benchmarking the biosensor against established methodologies, creating a user-friendly interface for utilization by non-specialists, and testing the device in the field to obtain feedback from users.

The present study describes a microfluidic chip for the immune-magnetic capture and immuno-electrochemical quantification of PSA [11,12]. This microfluidic immunosensor is coupled to a platform designed for smartphones and the inexperienced user, called the *Smartphone FAST-PSA tool.* Interestingly, this system can be readily adapted for the determination of other biomarkers by changing the working solutions (Figure 1A). In this study, the analytical parameters of this device were tested in reference samples, and it was used to measure PSA levels in 96 blood samples obtained at different time points from 50 donors with PCa, comparing findings with the results of gold standard methodologies.

## 2. Materials and Methods

### 2.1. Apparatus

Fabrication of the microfluidic chip and electrodes utilized a μPG 101 desk-top laser writer (Heildelberg Instruments, Heidelberg, Germany), Karl Suss MA6 Mask Aligner (Suss Microtec, Garching, Germany), Diener Asher RF plasma barrel reactor (Diener electronic, Ebhausen, Germany) and AJA ATC-1800 sputtering system (AJA International Inc., MA, USA). The Ammis ImageStream X Mk II imaging cytometer (Luminex, Austin, TX, USA) and Zeiss GEMINI high-resolution Scanning Electron Microscope (SEM) were employed to characterize the microbeads. A Sensit Smart Potentiostat (Palm Sens, Houten, The Netherlands) was used for amperometric detection.

### 2.2. Electrode Fabrication

The structured electrodes were fabricated as previously described [49]. Briefly, pre-stressed polystyrene sheets (PS) were washed with isopropanol, ethanol, and water, dried under nitrogen flow, and coated with a SU-8 photoresist (MicroChem, Newton, MA, USA) layer of 2 μm thickness. Then, self-adhesive vinyl was cut with the specific patterns and dimensions of the working (WE), auxiliary (AE), and reference (RE) electrodes and placed onto the PS-coated slide. Gold and platinum were deposited using the sputtering system at a deposition rate of ~1 Å/s (100 nm) for gold and ~0.1 Å/s (150 nm) for platinum. Next, the vinyl was removed, and the PS slides were placed in an oven at 160 °C to shrink the PS. Finally, the electrodes were lifted by dissolving the PS in an acetone bath and were conserved in acetone until further use.

### 2.3. PDMS Microfluidic Device Fabrication

The fabrication of this device was performed according to the protocol described by Saem et al. [50]. Briefly, The microfluidic pattern was designed with nanoCad (Nanosoft, London, UK) and drawn on glass coated with a photo-sensitive material to create the mask by soft lithography as described by Saem et al. [50]. The mask was then aligned over the SU-8 (photoresist)-coated silicon wafer and exposed to UV light in a μPG 101 desk-top laser writer to create the mold. Next, after washing the non-polymerized photoresist, the mold was placed on a petri dish and coated with PDMS prepolymer previously mixed with the curing agent (Down, Midland, TX, USA). Air bubbles were removed by exposing the molds to multiple vacuum cycles. The PDMS was cured at 80 °C for 3 h and then cut and peeled from the mold for cleaning and activation with O_2_ plasma. Finally, the upper slide with the microfluidic channel was pressed against the lower slide with the electrodes (Figure 1B).

### 2.4. Characterization of the Solid Support

Commercial Dynabeads-COOH (Invitrogen^TM^, Waltham, MA, USA) were purchased from Thermofisher Scientific. For characterization of their size, morphology, and concentration by imaging flow cytometry, the microbeads were diluted 1/100 in 10 mM phosphate-buffered saline (PBS) with pH of 7.20 at 40 mm/s, and each event was photographed at 60× magnification using ImageStream equipment. The IDEAS software was used for the data analysis. For characterization by SEM, the microbeads were fixed in a solution of PBS with 4% paraformaldehyde and 1% glutaraldehyde for 24 h and were then washed in PBS followed by incubation in 2% osmium tetroxide for 1 h. Microbeads were then fully dehydrated in serial ethanol solutions. Finally, the dry sample was sprinkled onto slide glass and metalized for photography with the Zeiss Gemini scanning electron microscope.

### 2.5. Immobilization of the Anti-PSA in Magnetic Microbeads

A suspension of homogenized commercial COOH microbeads (Thermo Scientific, Waltham, MA, USA) was washed twice with 0.1 M NaOH solution for 10 min and twice with H_2_O for 10 min. The beads were then magnetically separated to remove the supernatant, followed by activation of the surface carboxylic groups of the microbeads in a freshly prepared 1-Ethyl-3-(3-dimethylaminopropyl) carbodiimide (EDC) solution for 30 min. The microbeads were then washed twice with 2-(N-morpholino) ethanesulfonic acid (MES) buffer before incubation in a solution with excess PSA antibody (PSA-Ab) in MES buffer. Next, the microbeads were incubated in a 1 M ethanolamine solution to block residual activated groups. Finally, they were washed once with 0.1 M Tris-HCl buffer (pH 7.2) and twice with PBS. The functionalized microbeads were stored at 4 °C in PBS until their utilization.

### 2.6. Analytical Procedure for PSA Determination

All solutions were injected at a flow rate of 2 µL min^−1^. A schematic representation of this procedure is depicted in Figure 1A. Briefly, 1 × 10^6^ of PSA-Ab-MBs were dissolved in 50 μL of 1% BSA, 0.1% Tween, and 2% goat serum in PBS and incubated for 10 min. Next, 10 μL of circulating plasma samples were added and incubated in a shaker for 15 min at room temperature. The sample was then collected into a 0.1 mL syringe and manually injected into the central channel of the microfluidic system. Next, the beads were washed with 10 mM PBS pH 7.20 for 4 min, and HRP-conjugated anti-PSA (diluted 1000-fold with 10 mM PBS pH 7.20) was then added for a further 5 min, followed by another wash for 4 min. Finally, the enzyme substrate (1 mM H_2_O_2_ + 1 mM 4-TBC in 10 mM phosphate-citrate buffer pH 5.00) was injected, and the electrochemical reaction was detected at −100 mV by amperometry using the PStouch app for Android and PSTrace for Windows (Palm Sens, Houten, The Netherlands).

### 2.7. Blood Sample Collection

The study was approved by the Ethical Committee of Virgen de las Nieves University Hospital and complied with the principles of the Declaration of Helsinki. Written informed consent was obtained for all of the samples. The patients were enrolled and followed at the Urology and Oncology Departments of the hospital. Appendix A exhibits the clinical data of the study participants.

## 3. Results

### 3.1. Characterization and Functionalization of the Solid Support

Commercial Magnetic MBs Dynabeads^TM^ were characterized by image cytometry and SEM. A total of 1 × 10^4^ particles were photographed and analyzed by Image Cytometer ImageStream^TM^, revealing the circular and homogeneous beads depicted in Figure 2A,B. Image analysis using IDEAS software showed a mean circularity of 0.9898 ± 0.00549 (range 0.03502), a mean diameter of 2.787 ± 0.06618 μm (range 0.408 μm) and a mean area of 6.1 ± 0.2913 μm^2^ (range, 1.778 μm^2^). SEM images also depicted highly homogenous spherical particles with a diameter of around 2.7 μm and marked surface rugosity (Figure 2C).

The microbeads were functionalized with antibodies against human PSA, as described in the Methods section. The amount of bound microbead antibody was calculated by determining the initial and final anti-PSA antibody concentrations before and after the coupling reaction (Appendix A). An excess of antibody was observed in the initial solution and a reduction of 41.22 ± 0.13% after the coupling reaction (Appendix A).

### 3.2. Optimization of Experimental Variables

A control dilution of 800 pg mL was used for the optimization of experimental variables^−1^. The optimal flow rate was determined by evaluating the current intensity at different flow rates. As shown in Figure 3A, the flow rates from 1 to 2.5 μL min^−1^ had little effect on the signal obtained, which was markedly reduced at flow rates > 3 μL min^−1^. The optimal pH was then determined by testing solutions in a pH range of 4–7, observing the maximum current intensity at pH 5 (Figure 3B).

### 3.3. Quantitative Determination of PSA Biomarker in the Microfluidic Immunosensor

The *Smartphone FAST-PSA tool* platform was used to determine the current of serial PSA dilutions under the optimized conditions. The linear regression equation was i (nA) = 9.534 + 0.230 × CPSA, with linear regression coefficient of 0.997. The methodology shows a linear correlation of 10–1500 pg mL^−1^ (Figure 4A).

Intra-assay and inter-assay coefficients of variation (CVs) were then determined in solutions of 10, 800, and 1600 pg/mL^−1^ of PSA (five determinations in each), finding an intra-assay CV of 3.82% at 800 pg mL^−1^ and an inter-assay CV of 5.24% at 800 pg mL^−1^ (Table 1).

Next, a commercial ELISA test was used to plot the absorbance changes against the corresponding PSA concentration of serially diluted samples. The linear regression equation was A = 0.033 + 0.001 × CPSA, with a regression coefficient of 0.947 and CV of 6.47% for the determination of 800 pg mL^−1^ PSA (five replicates), a lower precision than obtained with the present platform.

The limit of detection (LOD) was defined as the lowest concentration yielding a signal three times the standard deviation of the blank. The LOD was 8 pg mL^−1^ for the ELISA test, higher (less sensitive) than the LOD of 2 pg mL^−1^ for the present platform.

The proposed analytical methodology was compared with a commercial ELISA, using both to analyze 15 samples with different PSA levels. The slopes obtained were close to 1 (r = 0.997), indicating a good correlation of the two methods (Figure 4B).

### 3.4. Clinical Performance of the Smartphone FAST-PSA Tool Platform for PSA Analysis

The specificity and selectivity of the platform to determine PSA in serum were tested by spiking a pool of serum from healthy donors with PSA at different dilutions. As shown in Figure 5A, the added PSA was fully correlated with the concentration measured by the sensor in the platform. Linear regression analysis showed a slope of 1.002 ± 0.0024 with 95% confidence interval (CI) of 0.9972 to 1.007 and slope intercept at 1.695 ± 0.0982 (95% CI, 1.492 to 1.899), the baseline concentration in the serum pool from the donors. Next, 96 liquid biopsy samples from 50 patients underwent different treatments at baseline (before treatment) and at 6 and 12 weeks. The results were also validated against the two gold standard methods routinely used in the hospital laboratory, RIA and ELISA, finding an excellent correlation (Figure 5B), with a linear regression slope of 1.050 ± 0.0108 (95% CI, 1.030 to 1.071).

A test was also conducted of the platform’s capacity to monitor the effects of therapy on PSA levels, which were determined in 22 patients with PCa at baseline and after 6 and 12 weeks of chemical castration treatment (Figure 5C). The PSA levels were markedly reduced by the treatment and were slightly lower after 6 versus 12 weeks of treatment.

## 4. Discussion

This study presents a novel microfluidic portable immunosensor-based method for the fast and reliable in situ quantification of PSA by clinicians using a smartphone.

The need to avoid any delay in the diagnosis of PCa has led to the development of rapid portable platforms that allow PSA levels to be determined in the clinical setting. Barbosa et al. [51] tested a fluoropolymer microfluidic device to quantify PSA using a smartphone and reported promising results, although the influence of ambient light on measurements limits its usefulness as a standardized methodology for application across centers with different lighting. This possibility of inter-assay error is markedly reduced by combining electrochemistry on a microfluidic chip. Mavrikou et al. [52] developed a biosensor whose membrane potential is modified by engineered cells in the presence of PSA, indicating a high (>4 ng mL^−1^) or low (<4 ng mL^−1^) PSA level. However, the use of cells as the sensing element limits the ease of use and the storage and portability of the device. For their part, Srinivasan et al. [53] coupled a gold nanoshell with anti-PSA antibody to obtain a colorimetric reaction on strip paper read in a custom cube but achieved only a semi-quantitative analysis. Numerous proposals for electrochemical PSA detection are described in a recent review [54]; however, the present device is the first PSA biosensor integrated within a fully portable platform that can be connected to a smartphone through the incorporation of a miniaturized potentiostat. The software for operating this device is highly intuitive and can be downloaded free from https://www.palmsens.com/software/pstouch/ (accessed on 11 June 2020), allowing measurements to be run, standard curves to be loaded and saved, and peaks to be analyzed and manipulated. It also permits the sharing of results via email or any other platform and supports other types of accessories (e.g., syringe pumps). Finally, the supplier allows for the modification and personalization of the interface using open code format.

There are certain limitations in the utilization of PSA as a biomarker of PCa, with reports of false positives and a resulting overdiagnosis and overtreatment of patients. However, the present platform can be simply adapted to detect other biomarkers by replacing the syringe content with different reaction components (Figure 1A). The system can even be modified on demand to include new biomarkers or other proteins found to be relevant for cancer diagnosis and treatment. Furthermore, the incorporation of functionalized magnetic microbeads offers high versatility, reduces incubation and washing times, and increases the reaction surface of the sputtered electrodes, thereby enhancing the sensitivity of the biosensor [39,44]. In summary, the device provides a reliable quantification of PSA that would be easy to implement at all heath-care levels.

## 5. Conclusions

The *Smartphone FAST-PSA tool* is a novel portable platform for smartphones based on the immunomagnetic immobilization of PSA and its immuno-electrochemical quantification. Its performance was tested in serial dilutions of PSA and in human samples, yielding the same values as those obtained by the gold standard methods. Advantages of this microfluidic immunosensor include its stability and its high selectivity and sensitivity, attributable to the immobilization of monoclonal antibodies by the magnetic microbeads. It provides the clinician with a rapid and reliable measurement of PSA levels and its fabrication is not costly, further supporting its potential for clinical implementation at all healthcare levels.

## Figures and Tables

**Figure 1 cancers-14-04483-f001:**
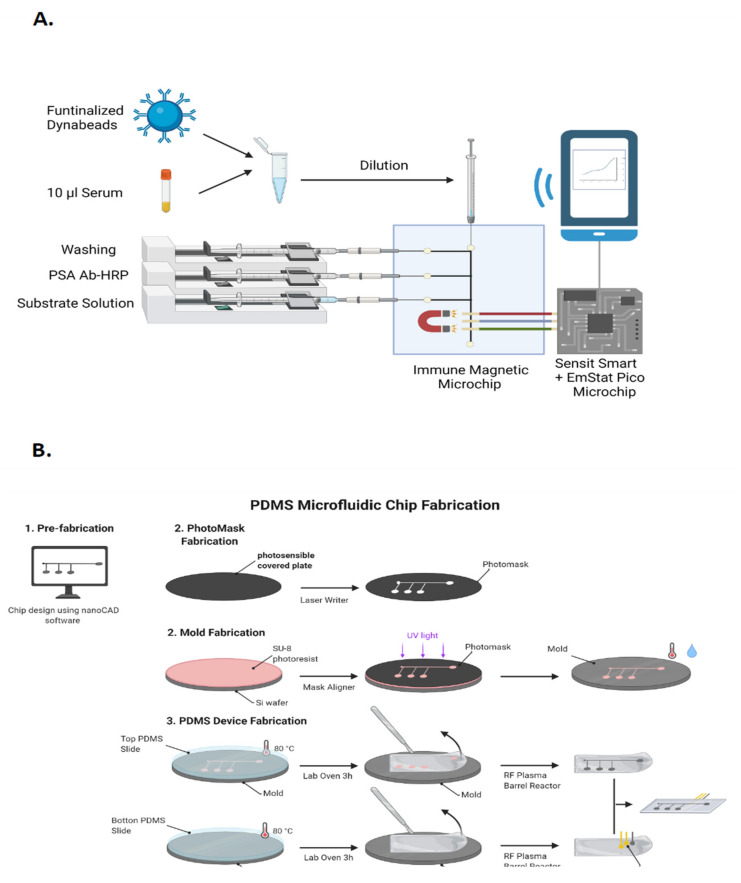
(**A**) Schematic representation of the *Smartphone FAST-PSA tool*. (**B**) Schematic representation of the microfluidic chip fabrication.

**Figure 2 cancers-14-04483-f002:**
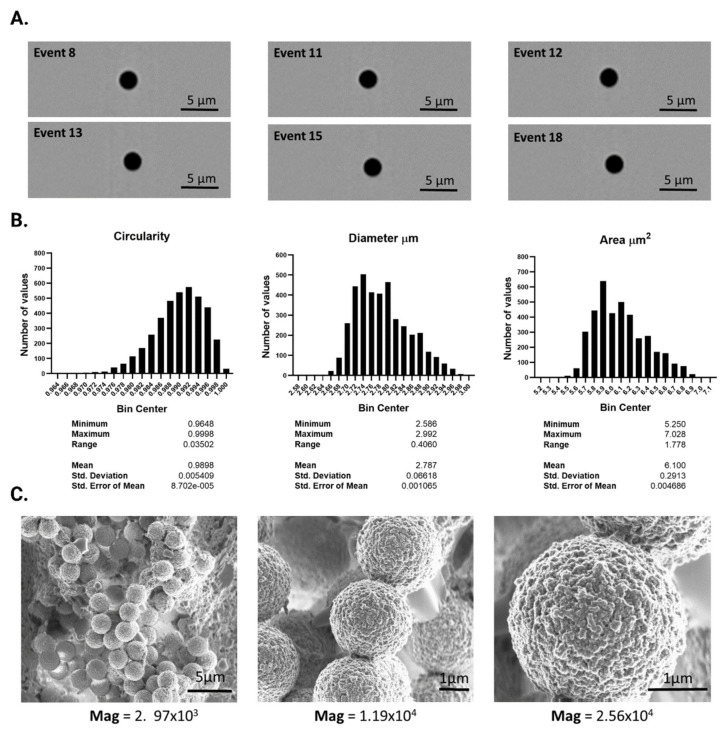
(**A**) Representative images obtained by image cytometry. (**B**) Histograms showing the frequency distribution of circularity, with perfect circularity = 1, diameter expressed in μm, and area expressed in μm^2^. (**C**) Representative SEM micrographs of the Dynabeads^TM^, obtained at several magnifications (Mag).

**Figure 3 cancers-14-04483-f003:**
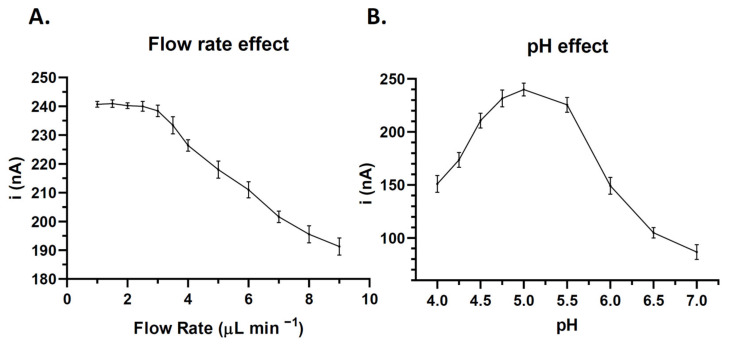
(**A**) Correlation between flow rates of electrochemical substrate dilution and the current generated on the electrode surface at −100 mV. Each dot represents the mean of 5 determinations and the error bar corresponds to the standard deviation. (**B**) Correlation between the pH of electrochemical substrate dilution and the current generated on the electrode surface at −100 mV. Each dot represents the mean of 5 determinations and the error bar corresponds to the standard deviation.

**Figure 4 cancers-14-04483-f004:**
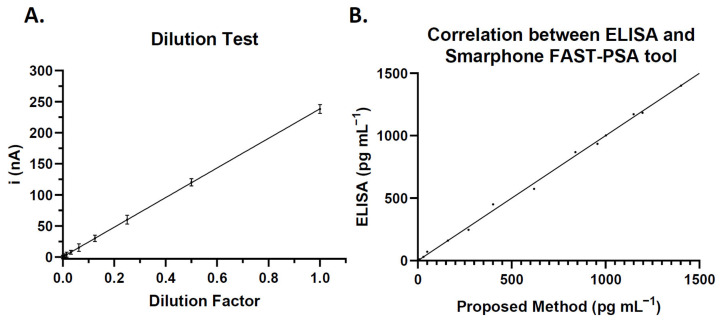
(**A**) Correlation between serial dilution of PSA in 10 mM PBS (pH 7.20) and the current level determined by the method. Each value is the mean of five determinations, and error bars represent standard deviations. (**B**) Correlation between *Smartphone FAST-PSA tool* platform and commercial ELISA.

**Figure 5 cancers-14-04483-f005:**
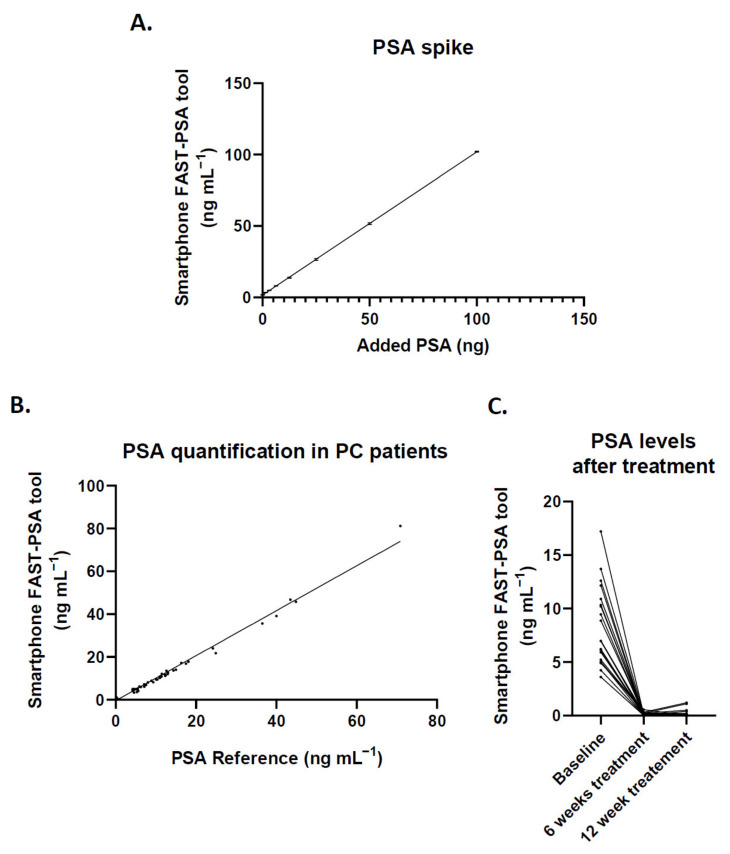
(**A**) The specificity and accuracy of the methodology were determined by spiking serum from healthy donors with known amounts of PSA; this graph depicts the mean of three independent experiments, with an error bar representing the standard deviation (SD). (**B**) Comparison of PSA data between the *Smartphone FAST-PSA tool* and laboratory testing (RIA or ELISA). Each dot represents the same sample analyzed by both methods. (**C**) Determination of the usefulness of the tool for the follow-up of patients with PCa; the graph depicts PSA levels at baseline (before treatment) and at 6 and 12 weeks of chemical castration treatment.

**Table 1 cancers-14-04483-t001:** Intra-assay precision (five measurements in the same run for each control sample) and inter-assay precision (five measurements for each control sample, repeated for three consecutive days).

^a^ Control Sample	Intra-Assay		Inter-Assay	
	Mean	CV%	Mean	CV%
10	9.98	2.71	10.04	4.46
800	799.95	3.82	800.02	5.42
1200	1199.98	3.50	1200.03	6.16

^a^ pg/mL PSA.

## Data Availability

The data presented in this study are available in the Appendix A and on request from the corresponding author.

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
