# Peer review of "A Novel, Quick, and Reliable Smartphone-Based Method for Serum PSA Quantification: Original Design of a Portable Microfluidic Immunosensor-Based System"

_cancers, 2022, doi:10.3390/cancers14184483_

Round 1

Reviewer 1 Report (Previous Reviewer 3)

The new and revised text addresses this reviewer's concerns about the initial draft.

Author Response

Author would like to thanks to reviewer 1 for his helpful comments.

English of the final version of the manuscript have been improved as reviewer 1 suggested.  

Reviewer 2 Report (Previous Reviewer 2)

The Authors have done a good job in revising their manuscript and improving presentation and scientific soundness. The pairs looks a lot better now

Author Response

Author would like to thanks to reviewer 2 for his helpful comments.

English of the final version of the manuscript have been improved as reviewer 2 suggested.  

Reviewer 3 Report (Previous Reviewer 1)

The reviewer thinks that the authors have not fully addressed the reviewer's previous comments.

Author Response

Author would like to thanks to reviewer 3 for his helpful comments. 

This manuscript is a resubmission of an earlier submission. The following is a list of the peer review reports and author responses from that submission.

Round 1

Reviewer 1 Report

Manuscript ID: cancers-1806520

Title: Portable Microfluidic Immunosensor for PSA Quantification, a Concept on Demand for Liquid Biopsy Application

Authors: Francisco Gabriel Ortega, German Gomez, Coral Gonzalez Martinez, Teresa Valero, Jose Exposito-Hernández, Ignacio Puche, Alba Rodriguez-Martinez, María Jose Serrano, José Antonio Lorente, Martin A. Fernández-Baldo*.

This manuscript reported a portable microfluidic device for PSA detection based on an electrochemical method with a miniaturized potentiostat. The reviewer thinks this manuscript has low novelty, low significance of content, and low quality of result presentation. The reviewer does not think this manuscript is of a sufficient standard to be accepted for publication in Cancers.

The reviewer thinks this approach is not novel at all, even if the authors described, To the best of our knowledge, the determination of PSA in liquid biopsy by means of this kind of portable platform with microfluidic immunosensor has not been previously reported”. There are numerous articles describing “the determination of PSA in liquid biopsy” and/with “portable platform” and/with “microfluidic immunosensor”. For example, the reviewer quickly searched in a minute, and two related research articles are found. [1] Biosensors (Basel). 2022 Apr 19;12(5):259. doi: 10.3390/bios12050259, [2] Biosens Bioelectron. 2022 Sep 1;211:114413. doi: 10.1016/j.bios.2022.114413. Besides these, tons of articles have been published with a similar concept as the reviewer knows.

I think the authors had already attempted to answer the above questions by following sentences (blue marked) in the discussion section of this manuscript. “During the last years some works have been published on this thematic: Barbosa et.al. have developed a fluoropolymer microfluidic device to quantify PSA by a capture with a smartphone with promising results but some issues to standardize the methodology. Mavrikou et.al. [48] have developed a biosensor where engineered cells modify its membrane potential in presence of PSA, corresponding with a high (>4 ng mL-297 1) and low (<4 ng mL-1) PSA level. Coupling a gold nanoshell with anti PSA antibody, Srinivasan et.al. [49] obtain a colorimetric reaction in a test line of a strip paper platform and read in a custom cube for PSA semi-quantification. According to a recent review [50] there is an important increase in the number of works related with electrochemical detection of PSA but, none of them are integrated in a portable platform or could be connected to a smartphone thanks to the incorporation of our miniaturized potentiostat.” However,

-        The authors should address, what are “some issues” in the research of “Barbosa et al. [Biosens Bioelectron. 2015 Aug 15;70:5-14. doi: 10.1016/j.bios.2015.03.006]? And, why do those “some issues” become a hurdle to clinical application, and how did this manuscript improve and make better the system for clinical application. In addition, “some issues” is totally not a scientific word.

-        Could the authors explain the drawback of the research of Mavrikou et al. [48] and Srinivasan et al. [49] in terms of applicability for clinical study or practice, compared to the author’s platform.

-        The reviewer does not think “integration in a portable platform or could be connected to a smartphone” is no longer meaningful or important, if either performance and user interface are not significantly better than other technologies. Could the authors objectively compare/evaluate the performance including user-interface of the author's platform to any other commercial products or published techniques, even if the other techniques do not adapt “electrochemical” and/or “potentiostats”. In addition, please refer to the below reference to understand the current trends in prostate cancer diagnostic devices. [3] Sensors (Basel). 2021 Jul 24;21(15):5023. doi: 10.3390/s21155023.

Reviewer 2 Report

This paper is interesting and brings some novelty into the Oncological Urology field. In particular, the possibily to obtain a quick and reliable evaluation of PSA level in patients with prostate diseases can be very promising and attractive for Urologists and Oncologists.

Several spell checks are required in the text, so the manuscript should be proof-read by a native speaker.

The paper's title can be significantly improved. I think the title of this paper should be more attention-catching, and should stimulate the curiosity of the readers. Moreover, it should be more comprehensible. For example, a good title could be: "A NOVEL, QUICK AND RELIABLE SMARTPHONE-BASED METHOD FOR SERUM PSA QUANTIFICATION: ORIGINAL DESIGN OF A PORTABLE MICROFLUIDIC IMMUNOSENSOR-BASED SYSTEM.

Simple Summary:

Line 21: "Prostate cancer is the most frequently diagnosed malignancy and the second leading cause of cancer-related death in male patients. Early diagnosed patients have better prognosis than those diagnosed in more advanced stages. Currently, Prostatic Specific Antigen (PSA) is the most used and analyzed  biomarker for prostate cancer...".

Line 29-30: re-phrase, unclear

Abstract:

Line 43-45: re-phrase, unclear

Introduction: 

Add to the references a paper about biomarkers use in Urology and prostate cancer, like Mancini M., et al: Stem cells, Biomarkers and Genetic Profiling: approaching future challenges in Urology, Urologia J, 2016, DOI:10.5301/uro/5000165.

Line 61-62: re-phrase, unclear

Line 66: PSA is influenced not only by prostate size and by androgen activity, but also and prominently by prostate inflammation. This concept should be added by the Authors.

Line 69-71: re-phrase, unclear

Line 78: eassy?

line 98-100: "To the best of our knowledge, the determination of serum PSA using this novel  kind of portable platform, based on a micro-fluidic immunosensor system,  has not been previously reported (this is the novelty and strenght of this paper, so it should be stressed)

Methods:

Line 128: petri dish.   

line 129: air bubbles

line 178: Patients were enrolled and followed at the Urology and Oncology Departments of our Hospital.

Results:

line 186: We observed remarkably circular and homegenous beads...

line 193: in the Methods section

Discussion:

line 285: describe clearly that you  developed and tested a novel microfluidic portable immunosensor-based method for fast and reliable quantification of PSA, and you propose it to clinicians in an office-based fashion, using a smartphone. This is a real new feature, which makes this paper very attractive to the readers (Urologists in particular), for its possible future clinical applications.

At the end of the Discussion, expand the concept that PSA is not an optimal biomarker, and can lead to false positives, overdiagnosis and overtreatment of patients with prostate cancer. However, your method is able to be updated, and can read several other biomarkers, just by modifying the content of the syringe with various reaction components. Add that the system could be even fine-tuned "on demand" to include new biomarkers or other proteins relevant for cancer diagnosis and treatment which will be discovered in the next future. This is definitively another strenght of your paper. 

Conclusions: 

You mention here the "FEPMA" platform. It was mentioned also in the Results section, line 223. Do you assume that the readers already know this platform? If not, shouldn't  you explain clearly what is it? If you have designed this platform, so this is an original work of you, shouldn't you choose a name that is easier to pronounce in English and easier to remember? If FEPMA stands for Fast Easy Protein Marker Analyzer, can't you find a less cacophonic acronym? Like for example Fast E-PROMA Wizard? Or Smartphone FAST-PSA tool? Think about it. To find an easy-to remember and friendly name is very important to introduce a new idea to the public.

Reviewer 3 Report

The authors describe a microfluidic, point-of-care device for PSA quantification in prostate cancer patient blood. The experiments are well-designed, however, the authors overstate the novelty of of their work.

Specifically, the assertion that "To the best of our knowledge, the determination of PSA in liquid biopsy by means of this kind of portable platform with microfluidic immunosensor has not been previously reported." Some highly-cited papers describing similar instrumentation include:

Triroj et al., Biosensors and Bioelectronics 2011 https://doi.org/10.1016/j.bios.2010.11.039

Chikkaveeraiah et al., Biosensors and Bioelectronics 2011 https://doi.org/10.1016/j.bios.2011.05.005

Zani et al., Electroanalysis 2010 https://doi.org/10.1002/elan.201000486

Tang et al., ACS Sensors 2016 https://doi.org/10.1021/acssensors.6b00256

The authors should use the introduction and discussion sections to better differentiate their technology from previously-published devices and explain the relative advantages.

The manuscript contains minor grammar and spelling errors throughout, particularly in terms of mixing singular and plural words in the same sentence. Some specific issues include:

In the simple summary “Prostate Cancer is one of the second leading cause….”

Abstract: “sensibility” should be “sensitivity”

Figure 1: “Functionalized” spelled wrong.

Figure 3, 4, and 5 captions: “the media of 5 determinations”. Is this supposed to be median or mean?

Section 2.5: the reagent EDC is not spelled out

Section 2.6: Wording related to HRP-anti-PSA addition, timing, and rinsing is unclear